

Environmental changes during the onset of the Late Pliensbachian
Event (Early Jurassic) in the Mochras Borehole, Cardigan Bay Basin,
NW Wales.
Teuntje P. Hollaar[1,2], Stephen P. Hesselbo[2,3], Jean-François Deconinck[4], Magret Damaschke[5],
Clemens V. Ullmann[2,3], Mengjie Jiang[2], Claire M. Belcher[1].
1 WildFIRE Lab, Global Systems Institute, University of Exeter, Exeter, EX4 4PS, UK
2 Camborne School of Mines, Department of Earth and Environmental Science, University of Exeter, Penryn
Campus, Penryn, TR10 9FE, UK
3 Environment and Sustainability Institute, University of Exeter, Penryn Campus, Penryn, TR10 9FE, UK
4 Biogéosciences, UMR 6282 CNRS, Université de Bourgogne/Franche-Comté, 21000 Dijon, France
5 Core Scanning Facility, British Geological Survey, Keyworth, NG12 5GG, UK
*Correspondence to*: Teuntje P. Hollaar (t.p.hollaar@exeter.ac.uk)
**Abstract.** The Late Pliensbachian Event (LPE), in the Early Jurassic, is associated with a perturbation in the
global carbon cycle (positive carbon isotope excursion (CIE) of ~ 2 ‰), cooling of ~5°C, and the deposition of
widespread regressive facies. Cooling during the Late Pliensbachian has been linked to enhanced organic matter
burial and/or disruption of thermohaline ocean circulation due to North Sea doming. Orbital forcing had a strong
influence on the Pliensbachian environments and recent studies show that the terrestrial realm and the marine
realm in and around the Cardigan Bay Basin were strongly influenced by orbital climate forcing. In the present
study we build on the previously published data for long eccentricity cycle E459 ± 1 and extend the
palaeoenvironmental record to include E458 ± 1. We explore the environmental and depositional changes on
orbital time scales for the Mochras core during the onset of the LPE. Clay mineralogy, XRF elemental analysis,
isotope ratio mass spectrometry, and palynology are combined to resolve systematic changes in erosion,
weathering, fire, grain size and riverine influx. Our results indicate distinctively different environments before
and after the onset of the LPE positive CIE, and show increased physical erosion relative to chemical
weathering. We also identify 5 swings in the climate, in tandem with the 405 kyr eccentricity minima and
maxima. Eccentricity maxima are linked to precessionally repeated occurrences of a semi-arid, monsoonal
climate with high fire activity and relatively coarser fraction of terrestrial runoff. In contrast, 405 kyr minima in
the Mochras core are linked to a more persistent, annually wet climate, low fire activity, and relatively finer
grained deposits across multiple precession cycles. The onset of the LPE +ve CIE did not impact the expression
of the 405 kyr in the proxy records, however, during the second pulse of lighter carbon ($^{12}$C) enrichment, the
clay minerals record a change from dominant chemical weathering to dominant physical erosion.

### 1.1 Introduction

The Early Jurassic is a period marked by large climatic fluctuations and associated carbon-isotope excursions
(CIE's) in an overall warm and high $p$CO$_2$ world (McElwain *et al*., 2005; Korte & Hesselbo, 2011). A series of
small and medium sized CIE's have recently been documented for the Sinemurian and Pliensbachian, which
have mainly been recorded in European records (Korte & Hesselbo, 2011; Franceschi *et al*., 2014; Korte *et al*.,



2015; Price *et al.*, 2016; Hesselbo *et al.*, 2020a; Storm *et al.*, 2020; Silva *et al.*, 2021; Cifer *et al.*, 2022) and
recently at the NW end of the Tethys Ocean in Morocco (Mercuzot *et al.*, 2020) and in North America (De Lena
*et al.*, 2019). Notable is the pronounced positive CIE in the Late Pliensbachian, which has been called the Late
Pliensbachian Event (LPE) and is linked to climatic cooling (Hesselbo & Korte, 2011; Korte *et al.*, 2015) and a
supra-regional/global sea level low stand (Hallam, 1981; de Graciansky *et al.*, 1998; Hesselbo & Jenkyns, 1998;
Hesselbo, 2008). The LPE has been recognized by a positive shift in benthic marine oxygen-isotopes (~1.5–2
per mil) (Bailey *et al.*, 2003; Rosales *et al.*, 2004,2006; Suan *et al.*, 2010; Dera *et al.*, 2011a; Korte & Hesselbo,
2011; Gómez *et al.*, 2016; Alberti *et al.*, 2019, 2021), coeval with a positive shift in marine and terrestrial
carbon isotopes (~2 per mil) (Jenkyns & Clayton, 1986; McArthur *et al.*, 2000; Morettini *et al.*, 2002; Quesada
*et al.*, 2005; Rosales *et al.*, 2006; Suan *et al.*, 2010; Korte & Hesselbo, 2011; Silva *et al.*, 2011; Gómez *et al.*,
2016; De Lena *et al.*, 2019).
A cooler Late Pliensbachian climate has been suggested based on low $pCO_2$ values inferred by leaf stomatal
index data from eastern Australia (Steinthorsdottir & Vajda, 2015), the presence of glendonites in northern
Siberia (Kaplan, 1978; Price, 1999; Rogov & Zakharov, 2010), vegetation shifts from a diverse flora of different
plant groups to one mainly dominated by bryophytes in Siberia (Ilyina, 1985; Zakharov *et al.*, 2006), and
possible ice rafted debris found in Siberia (Price, 1999; Suan *et al.*, 2011). Whilst the presence of ice sheets is
strongly debated, a general cooling period (~5°C lower; Korte *et al.*, 2015; Gómez *et al.*, 2016) is evident from
several temperature reconstructions of NW Europe. A cooling is hypothesized via enhanced carbon burial in the
marine sediments, leading to lower $pCO_2$ values and initiating cooler climatic conditions (Jenkyns & Clayton,
1986; Suan *et al.*, 2010; Silva *et al.*, 2011; Storm *et al.*, 2020). Direct evidence of large-scale carbon burial in
Upper Pliensbachian marine deposits has not yet been documented (Silva *et al.*, 2021). Alternatively, cooling
has been suggested to be caused by regional tectonic updoming of the North Sea region, which would have
disrupted the ocean circulation in the Laurasian Seaway, reducing poleward heat transport from the tropics
(Korte *et al.*, 2015). Disruption of the ocean circulation between the western Tethys and the Boreal realm is
supported by marine migration patterns (Schweigert, 2005; Zakharov *et al.*, 2006; Bourillot *et al.*, 2008;
Nikitenko, 2008; Dera *et al.*, 2011b; van de Schootbrugge *et al.*, 2019) and numerical models (Bjerrum *et al.*,
2001; Dera & Donnadieu, 2012; Ruvalcaba Baroni *et al.* 2018); however, the direction of the flows remain
debated.
An additional factor at this time is that a strong orbital control exists on the sedimentary successions in the
Pliensbachian (Weedon & Jenkyns, 1990; Ruhl *et al.*, 2016; Hinnov *et al.*, 2018; Storm *et al.*, 2020; Hollaar *et*
*al.*, 2021). Previous studies have indicated that sea level changes, possibly coupled to glacio-eustatic rise and
fall, occurred during the LPE on a 100 kyr (short eccentricity) time scale (Korte & Hesselbo, 2011). A high-
resolution record of charcoal, clay mineralogy, bulk-organic carbon-isotopes, TOC and $CaCO_3$ encompassing
approximately one 405 kyr cycle from the Llanbedr (Mochras Farm) borehole, Cardigan Bay Basin, NW Wales,
UK suggested that the long-eccentricity orbital cycle had a significant effect on background climatic and
environmental change at this time, particularly affecting the hydrological regime of the region (Hollaar *et al.*,
2021). This previous research focussed on orbital forcing of environmental change for a time lacking any large
excursion in $\delta^{13}C_{org}$, and so unaffected by perturbations to the global carbon cycle. Here, we expand on the
record of Hollaar *et al.* (2021) to cover two long eccentricity cycles (which we identify as cycle E459 ± 1 and



E458 ± 1 of Laskar *et al*. 2011), where the final parts of E458 and the start of E457 are interrupted by onset of
the Late Pliensbachian Event. We find that the long eccentricity forcing continued to dictate the precise timing
of major environmental changes in the Cardigan Bay Basin, including the initial step of the positive carbon
isotope excursion.

**1.2 Material**
**1.2.1 Palaeo-location and setting**
Associated with the break-up of Pangea, connections between oceans via epicontinental seaways were
established during the Early Jurassic, such as the Hispanic Corridor, which connected the north-western Tethys
and the eastern Panthalassa, and the Viking Corridor which linked the north-western Tethys Ocean to the Boreal
Sea (Sellwood & Jenkyns, 1975; Smith *et al.*, 1983; Bjerrum *et al*., 2001; Damborenea *et al*., 2012). The linking
passage of the NW Tethys Ocean and the Boreal Sea (south of the Viking Corridor) is the palaeogeographical
location of the Llanbedr (Mochras Farm) borehole, Cardigan Bay Basin, NW Wales, UK (Fig. 1) – referred to
hereafter as Mochras. Due to the location of the Mochras succession during the Late Pliensbachian, it was
subject to both polar and equatorial influences allowing the study of variations in the circulation in the N-S
Laurasian Seaway (including the Viking Corridor) prior to and across the LPE. Mochras was located at a mid-
palaeolatitude of ~35° N (Torsvik & Cocks, 2017).





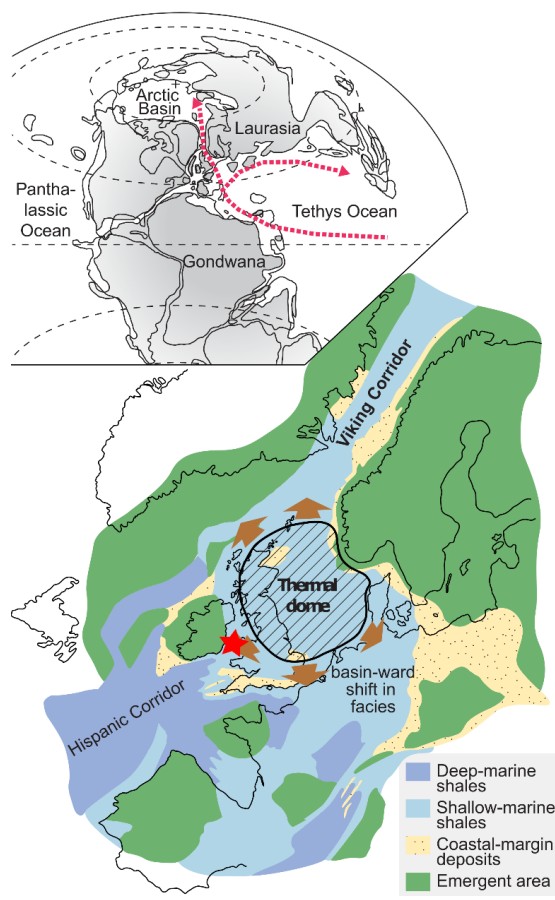


**Figure 1: Palaeolocation of the Mochras borehole in the context of potential North Sea doming.** Figure

reprinted and adapted from Korte *et al*. (2015), which is open access

(https://creativecommons.org/licenses/by/4.0/). The Mochras borehole was located at a paleolatitude of ~35° N

in the Cardigan Bay Basin (Torsvik & Cocks, 2017). Circulation in the Tethys Ocean and between there and the

Boreal region influenced the depositional environment of the Mochras core (Pieńkowski *et al*., 2021). Late

Pliensbachian uplift of the North Sea dome potentially led to occlusion of the Viking Corridor and disrupted

circulation in the seaway (Korte *et al*., 2015).

105

The depositional environment of Mochras is likely characterized by a rift setting, which is reflected by the

relatively open and deep marine facies and the evidence for below storm wave-base and contourite deposition

(Pieńkowski *et al*., 2021), but always with a strong terrestrial influence (van de Schootbrugge *et al*., 2005;

Riding *et al*., 2013) from the nearby landmasses (Dobson & Whittington, 1987). The Cardigan Bay Basin fill

was downthrown against the Early Paleozoic Welsh Massif on the SE side by a major normal fault system,

probably comprising the Bala, Mochras and Tonfanau faults at the eastern and south-eastern margins of the

basin in Late Paleozoic–Early Mesozoic time (Woodland, 1971; Tappin *et al*., 1994). The main source of



detrital material is understood to be the Welsh Massif, followed by the Irish Massif (Deconinck *et al*., 2019).
Other Variscan massifs that could have influenced the provenance are the London-Brabant Massif to the south
east, and Cornubia to the south (van de Schootbrugge *et al*., 2005), depending on the marine circulation and
sediment transport at the time.

### 1.2.2 Core location and material

The Llanbedr (Mochras Farm) Borehole was drilled onshore in the Cardigan Bay Basin (52 48' 32"N, 4 08'
44"W) in 1967–1969, North Wales (Woodland, 1971; Hesselbo *et al*., 2013). The borehole recovered a 1300 m
thick Early Jurassic sequence (601.83 – 1906.78 metres below surface (mbs)), yielding the most complete and
extended Early Jurassic succession in the UK, being double the thickness of same age strata in other UK cores
and outcrops (Hesselbo *et al*., 2013; Ruhl *et al*., 2016). The Lower Jurassic is biostratigraphically complete at
the zonal level (Ivimey-Cook, 1971; Copestake & Johnson, 2014), with the top truncated and unconformably
overlain by Cenozoic strata (Woodland, 1971; Dobson & Whittington, 1987; Tappin *et al*., 1994; Hesselbo *et*
*al*., 2013). The lithology is dominated by argillaceous sediments, with alternating muddy limestone, marl and
mudstone (Woodland, 1971; Sellwood & Jenkyns, 1975).

### 1.2.3 Pliensbachian

The Pliensbachian Stage in the Mochras borehole occurs between ~865 to ~1250 mbs, with the Margaritatus
Zone between ~1013 and 909 mbs (Page in Copestake & Johnson, 2014). The Pliensbachian interval comprises
alternations of mudstone (with a relatively moderate total organic carbon [TOC]) and organic poor limestones,
with a pronounced cyclicity at ~1 ± 0.5 m wavelength (Ruhl *et al*., 2016). The Upper Pliensbachian contains
intervals that are silty and locally sandy, whilst levels of relative organic enrichment also occur through the
Pliensbachian (Ruhl *et al*., 2016). Overall, the Upper Pliensbachian is relatively rich in carbonate (Ruhl *et al*.,
2016; Ullmann *et al*., 2022).

### 1.3 Methods

For this study, samples were taken at a ~30 cm resolution from slabbed core from 918–934 mbs for XRD and
mass spectrometry, as well as palynofacies and microcharcoal analysis. These new samples complement a set at
10 cm resolution from 951–934 mbs (results published in Hollaar *et al*., 2021).

### 1.3.1 TOC, CaCO₃ and bulk organic carbon isotope mass spectrometry

TOC and $\delta^{13}C_{org}$ were measured to track the changes in the total organic fraction and the bulk organic carbon
isotope ratios simultaneously with the other palaeoenvironmental proxy data.
Powdered bulk-rock samples (~ 2 g) were decarbonated in 50 ml of 3.3% HCl. After this, the samples were
transferred to a hot bath of 79 °C for 1 h to remove siderite and dolomite. Subsequently, the samples were
centrifuged and the liquid decanted. The samples were rinsed repeatedly with distilled water to reach neutral *p*H.
After this, the samples were oven-dried at 40 °C, re-powdered and weighed into tin capsules for mass
spectrometry with the Sercon Integra 2 stable isotope analyser at the University of Exeter Environmental &
Sustainability Institute (ESI), stable isotope facility on the Penryn Campus, Cornwall. Samples were run
alongside in house reference material (bovine liver; $\delta^{13}C$ -28.61 and Alanine; $\delta^{13}C$ -19.62) which was used to
correct for instrument drift and to determine the $\delta^{13}C$ values of the samples. $\delta^{13}C_{org}$ values reported relative to V-



PDB following a within-run laboratory standard calibration. Total organic carbon was determined using the $CO_2$ beam area relative to the bovine liver standard (%C 47.24). Replicate analysis of the in-house standards gave a precision of ± <0.1 ‰ (2 SD).

The carbonate content was measured by the dry weight sample loss before and after decarbonation. The %C content derived from the mass spectrometer was corrected for carbonate loss to derive TOC.

**1.3.2 X-Ray Diffraction (XRD) to determine clay mineralogy**

Clay mineral analysis was performed to gain insight into the relative importance of physical erosion versus chemical weathering and related changes in the hydrological cycle.

About 2–3 g of gently powdered bulk-rock was decarbonated with a 0.2 M HCl solution and the clay sized fraction (< 2 µm) extracted and oriented on glass slides for X-ray diffraction analysis (XRD) using a Bruker D4 Endeavour diffractometer (Bruker, Billerica, MA, USA) with Cu Kα radiations, LynxEye detector and Ni filter under 40 kV voltage and 25 mA intensity (Biogéosciences Laboratory, Université Bourgogne/Franche-Comté, Dijon). Following Moore & Reynolds (1997), the clay phases were discriminated in three runs per sample: (1) air-drying at room temperature; (2) ethylene-glycol solvation during 24 h under vacuum; (3) heating at 490 °C during 2 h.

Identification of the clay minerals was based on their main diffraction peaks and by comparing the three diffractograms obtained. The proportion of each clay mineral on glycolated diffractograms was measured using the MACDIFF 4.2.5. software (Petschick, 2000). Identification of the clay minerals follows the methods in Deconinck *et al*. (2019) and Moore & Reynolds (1997).

**1.3.3 Palynofacies and microcharcoal**

Palynofacies were examined to explore shifts in the terrestrial versus marine origin of the particulate organic matter. Each ~ 20 g bulk rock sample was split into 0.5 cm$^3$ fragments, minimizing breakage of charcoal and other particles, to optimize the surface area for extraction of organic matter using a palynological acid maceration technique. The samples were first treated with cold hydrochloric acid (10% and 37% HCl) to remove carbonates. Following, hydrofluoric acid (40% HF) was added to the samples to remove silicates. Carbonate precipitation was prevented, by adding cold concentrated HCl (37%) after 48 h. The samples were neutralized via multiple DI water dilution-settling-decanting cycles, after which 5 droplets of the mixed residue were taken for the analysis of palynofacies prior to sieving. The remaining residue was sieved through a 125 µm and 10 µm mesh to extract the microcharcoal fraction.

A known quantity (125 µl) of the 10–125 µm sieved residue was mounted onto a palynological slide using glycerine jelly. This fraction, containing the microscopic charcoal, was analysed and the charcoal particles counted using an Olympus (BX53) transmitted light microscope (40x10 magnification). For each palynological slide four transects (two transects in the middle and one on the left and right side of the coverslip) were followed and the number of charcoal particles determined. Charcoal particles were identified with the following criteria: opaque and black, often elongated lath-like shape with sharp edges, original anatomy preserved, brittle appearance with a lustrous shine (Scott, 2010). These data were then scaled up to the known quantity of the sample according the method of Belcher *et al*. (2005).



Palynofacies were grouped broadly according to Oboh-Ikuenobe *et al.* (2005): sporomorphs, fungal remains,
freshwater algae, marine palynomorphs, structured phytoclasts, unstructured phytoclasts, black debris,
amorphous organic matter (AOM), and charcoal (further described in Hollaar *et al*., 2021). The palynofacies
were quantified on a palynological slide using the optical light microscope (40x10 magnification) and counting
a minimum of 300 particles per slide. Because the samples are AOM-dominated, counting was continued until a
minimum of 100 non-AOM particles were observed. We used the percentage of terrestrial phytoclasts, which
includes sporomorphs, structured and unstructured phytoclasts, to examine changes in terrestrial organic particle
content.

**1.3.4 X-Ray Fluorescence (XRF) to determine detrital elements**
Detrital elemental ratios were examined to analyse changes in relative terrestrial influx and the type of material
transported from the land to the marine realm. The slabbed archive halves of the Mochras borehole were
scanned via automated X-ray fluorescence (XRF) at a 1 cm resolution for the interval 951 – 918 mbs, with the
ITRAX MC at the British Geological Survey Core Scanning Facility (CSF), Keyworth, UK (Damaschke *et al*.,
2021). The measurement window was 10 s and long-term drift in the measurement values was counteracted by
regular internal calibration with a glass reference (NIST-610). Duplicate measurements were taken every 5 m
for a 50 cm interval to additionally verify the measured results.
**1.3.5 Statistical analysis**
Principal component analysis (PCA) was performed in the software PAST on the normalized dataset including
microcharcoal, TOC, $CaCO_3$, $\delta^{13}C_{org}$, S/I, K/I, primary clay mineralogy, Si/Al, Zr/Rb. The samples before the
+ve CIE (951.0–930.4 mbs) and the samples after the +ve CIE (930.3–918.0 mbs) are grouped to examine a
potential difference in the sedimentary composition before and after the +ve CIE.
A Pearson's correlation was executed in Matlab R2017b. The *p* value tests the hypothesis of no correlation
against the alternative hypothesis of a positive or negative correlation (significance level at $p \leq 0.05$).

**1.4 Results**

**1.4.1 TOC, $CaCO_3$ and bulk organic carbon isotope ratio mass spectrometry**
Alternating TOC-enhanced and Ca-rich lithological couplets occur on a metre scale through the studied interval
(r = -0.64, p = 0.001). TOC content fluctuates in the range 0.17–1.72 wt% (mean 0.8 wt%) and the highest
fluctuations of TOC content are found from 939–930 mbs. The $CaCO_3$ content fluctuates in opposition of TOC
and varies between 14 and 89 %. The studied interval is generally high in $CaCO_3$ (mean 58 %) (Fig. 2). The
$\delta^{13}C_{org}$ displays a minor (~0.5 ‰) shift towards more positive values at ~944 mbs (as reported in Storm *et al*.,
2020; Hollaar *et al*., 2021). At ~ 930 mbs an abrupt shift of ~1.8 ‰ (Fig. 3 and 4; Storm *et al*., 2020) indicates
the onset of the Late Pliensbachian Event (LPE) in the Mochras core. In agreement with this, the results of the
present study show a shift from ~ minus 27 per mil to ~ minus 25.15 per mil between 930.8 and 930.4 mbs (Fig.
3). The $\delta^{13}C_{org}$ data presented here have been divided into three phases: the pre-LPE gradual rise, followed by
the +ve CIE, which is subdivided into pulses 1, 2 and 3 (Fig. 4). After the onset of the positive $\delta^{13}C_{org}$ excursion,
the TOC content drops to the lowest values (from 0.85 % before and 0.6 % after the +ve CIE on average), but



the 1 metre fluctuations continue (Fig. 2 and Fig. 3). No overall change in the CaCO₃ content is observed
through the positive carbon-isotope excursion (Fig. 2).

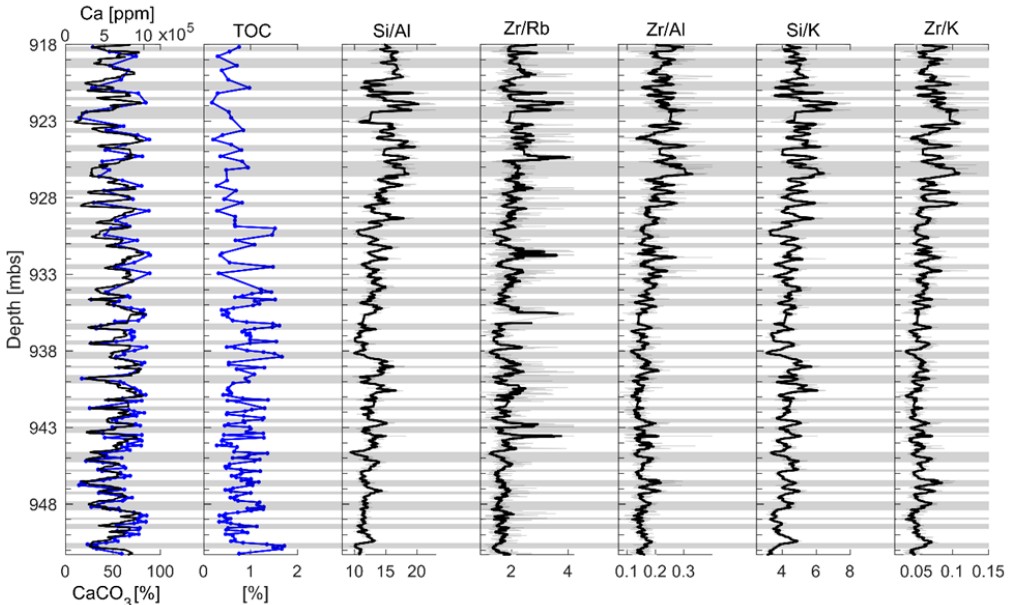


**Figure 2: Detrital ratios over the lithological Ca-rich and TOC-enhanced couplets for the studied**
**interval.** Overview of Ca (black, derived from Ruhl *et al*. 2016), CaCO₃ (blue), and TOC content of the studied
interval 951–918 mbs. The grey shading represents the TOC-enhanced lithological beds and the unshaded bands
mark the Ca-rich (limestone) beds. The detrital ratios reflect the silt to fine sand fraction (Si, Zr) versus the clay
fraction (Rb, Al, K). Two increasing upward cycles are observed in the Si/Al and Zr/Rb ratios. The pattern
observed in all detrital ratios (except the Ti/Al) is similar and likley reflects changes in grain size.

**1.4.2 Clay minerals**
XRD analysis shows that the main clay types found in this interval are illite, random illite-smectite mixed-layers
(I-S R0) [hereafter referred to as smectite], and kaolinite. Illite and kaolinite co-fluctuate in the interval studied
here, and are directly out of phase with smectite abundance. Chlorite and R1 I-S are present in minor
proportions, but reach sporadically higher relative abundance (> 10 %) from ~ 932 mbs upwards, with sustained
>10% abundance at ~925–918 mbs (Fig. 3 and SI Fig. 1). The relative abundances of smectite and illite and of
kaolinite and illite are expressed by the ratio S/I and K/I respectively. These ratios were calculated according to
the intensity of the main diffraction peak of each mineral.





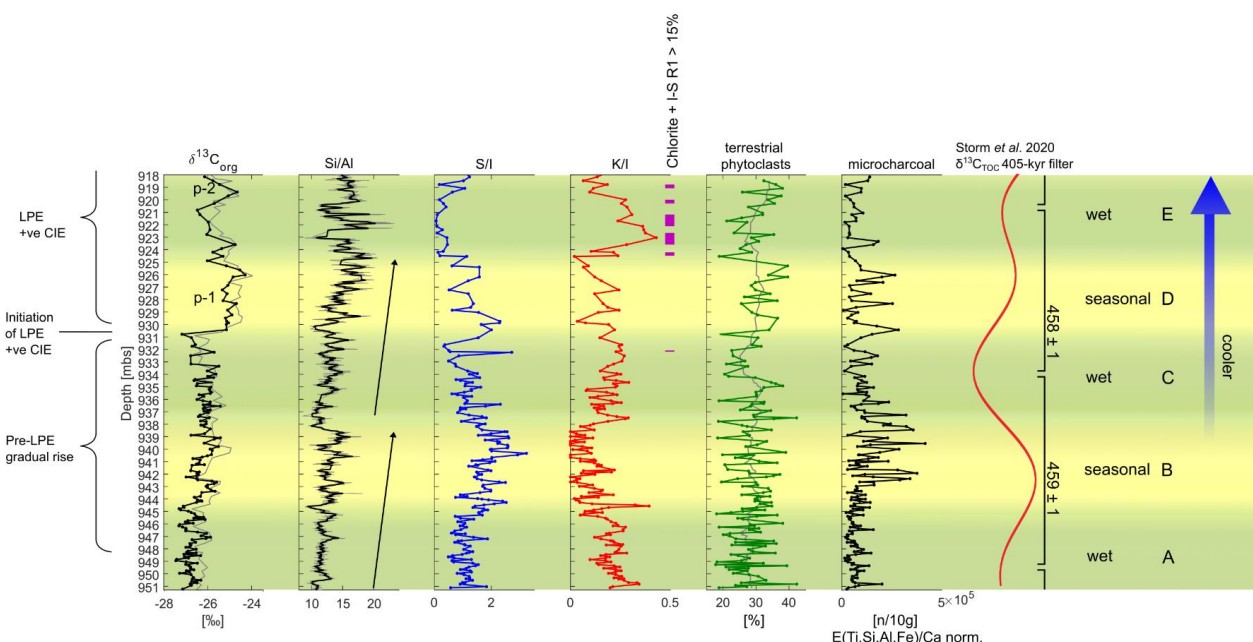


**Figure 3**: **Synthesis diagram showing the climatic swings observed in tandem with the long eccentricity**
**cycle.** The studied interval comprises part of the pre LPE gradual rise, the initiation of the LPE +ve CIE and
pulse 1 and 2. Five climatic phases (A–E) are interpreted from the Si/Al, smectite/illite, kaolinite/illite, chlorite
and I-S R1 abundance and the microcharcoal abundance. In tandem with the 405 kyr cycle (Storm *et al.*, 2020)
climatic state of a year-round wet climate, low fire activity and fine-grained sediments across multiple
precession cycles (phase A and C) alternates with a climatic state that includes repeated precessionally driven
states that are semi-arid, with high fire activity and coarser sediments (phase B and D). The top of the record
(phase E) indicates increased physical erosion (chlorite + I-S R1, kaolinite) relative to chemical weathering.












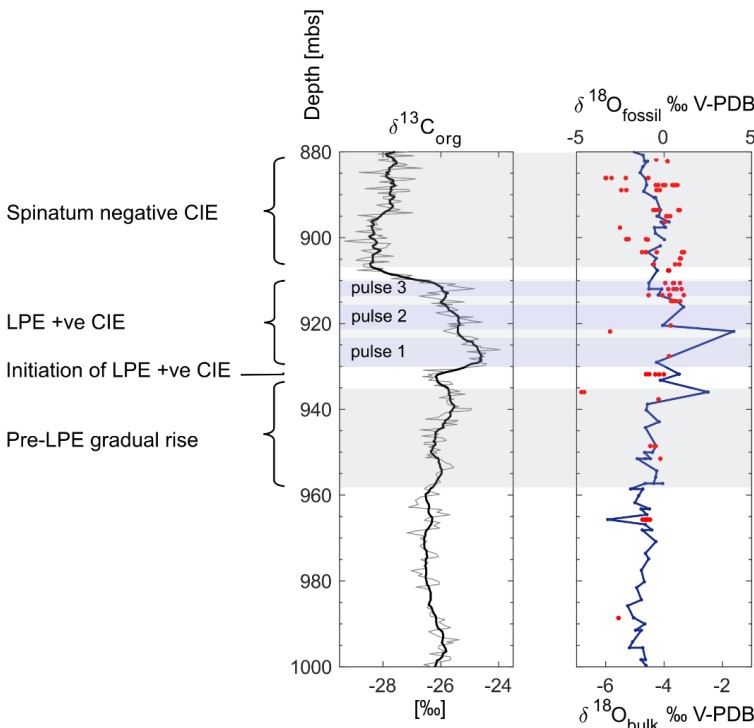


**Figure 4: The δ¹³C_org (Storm *et al*., 2020) and δ¹⁸O_bulk and δ¹⁸O_fossil (Ullmann *et al*., 2022) from the Late**
**Pliensbachian of the Mochras core.** A pre-LPE gradual rise is recorded in the $\delta^{13}C_{org}$ of the Mochras core,
followed by the initiation of the LPE +ve CIE, which consists of three pulses. After the LPE +ve CIE, $\delta^{13}C_{org}$
values drop and the Spinatum negative CIE is recorded. The $\delta^{18}O_{bulk}$ of the Mochras core (blue) are
diagenetically altered and are unlikely to preserve a palaeoclimatic imprint (Ullman *et al*., 2022); however, peak
values in the $\delta^{18}O_{bulk}$ occur during the LPE +ve CIE. Also, the $\delta^{18}O_{fossil}$ values (red) are slightly more positive
during pulse 3 of the LPE +ve CIE.

**1.4.3 Organic matter**
The type of particulate organic matter, and more specifically the abundance in the either marine or terrestrial
origin of the particles, fluctuate on a metre scale from 18–42 %. Palynofacies indicate that the type of organic
matter does not change in relation to the metre-scale lithological facies cycles (no correlation between
percentage terrestrial phytoclasts and TOC or $CaCO_3$). The proportion of terrestrial phytoclasts increases
towards the top of the record and has 4 high phases: between 944.6 and 942.0 mbs, 937.5 and 934.9 mbs, 930.4
and 925.4 mbs, and 920.3 and 918.0 mbs (SI Fig. 2). The first and second high phase falls within the + 0.5 ‰
positive swing in the $\delta^{13}C_{org}$; the latter two high phases correspond to pulse 1 and pulse 2 in the +ve CIE.





Amorphous organic matter (AOM) is very abundant, followed by unstructured phytoclasts, with lower
proportions of structured phytoclasts and charcoal (SI Fig. 3). Microcharcoal particles make up a relatively large
proportion of the terrestrial particulate organic matter (~10 % on average) and ~3.5 % on average of the total
particulate organic matter fraction (SI Fig. 3). Only sparse marine and terrestrial palynomorphs were observed
(SI Fig. 3). No abrupt changes are recorded in the terrestrial/marine proportions, but small long-term
fluctuations are observed in the percentage of terrestrial phytoclasts, with three phases of increase noted, of
which the overall highest phase occurs after the start of the +ve CIE.
To assess the character of the observed fluctuations in charcoal abundance, whether changes in charcoal can be
related to enhanced runoff from the land and/or organic preservation, or if the charcoal signifies changes in fire
activity on land, the charcoal record was corrected for detrital influx. We adjust the charcoal particle abundances
using the XRF elemental record, normalizing to the total terrigenous influx following Daniau *et al*. (2013) and
Hollaar *et al*. (2021). The stratigraphic trends in the normalized microcharcoal for $E_{ter}/Ca$, Si/Al, Ti/Al and
Fe/Al remain the same (SI Fig. 4). The absolute number of charcoal particles decreases, with raw mean charcoal
particles $1.06 \times 10^5$ per 10 g and $E_{ter}/Ca$ normalized mean $9.7 \times 10^4$ n/10g, Ti/Al normalized $6.4 \times 10^4$ n/10g, Si/Al
normalized $7.7 \times 10^4$ n/10g, Fe/Al normalized $9.8 \times 10^4$ mean number of microcharcoal particles per 10 g (SI Fig.
4). The number of charcoal particles per 10 g processed rock decreases when correcting for terrestrial run-off
changes, hinting that perhaps part of the 'background' charcoal is related to terrestrial influx; the normalisation
also shows that the observed patterns in charcoal abundances are not influenced by changes in terrestrial runoff
and taphonomy. Hence, the highs and lows in the charcoal record can be interpreted to represent changes in the
fire regime on land. The microcharcoal abundance fluctuates strongly in the record presented here; however, no
clear difference in charcoal content has been observed before and after the onset of the +ve CIE.

**1.4.4 Detrital elemental ratios (XRF)**
Strong similarities are observed between the fluctuating ratios of Si/Al, Si/K, Zr/Rb, Zr/Al and Zr/K (Fig. 2).
The elements Al, Rb and K sit in the clay fraction (e.g. Calvert & Pederson, 2007), whereas Si and Zr are often
found in the coarser fraction related to silt and sand grade quartz and heavy minerals (Calvert & Pederson,
2007). The ratios all show clear metre-scale fluctuations, and these are superimposed on two increasing-upward
trends observed in both the Si/Al and the Zr/Rb, followed by a drop and rise to peak values in the latest part of
phase D and phase E above the onset of the +ve CIE (Fig. 2; Fig. 3). A parallel trend is observed between the
clay ratios (XRD) and elemental ratios Si/Al and Zr/Rb (Fig. 2). Phases of high S/I correspond to the peaks in
the two coarsening upward sequences, whereas phases of high K/I correspond to the low phases in the two
coarsening upward sequences. After the +ve CIE onset (in phase E) this relationship turns around, and an
enrichment in the kaolinite/illite ratio corresponds to the elemental ratios, where highest kaolinite relative
abundance is observed in parallel with elemental ratios suggesting maximum coarse fraction.

**1.4.5 PCA analysis**
The proxy datasets ($\delta^{13}C_{org}$, TOC, percentage terrestrial phytoclasts, microcharcoal, smectite/illite,
kaolinite/illite, abundance of chlorite and R1 I-S, Si/Al, Zr/Rb, Zr/Al) were normalized between 0-1 and run for



PCA analysis in PAST. Sixty-four percent of the variance is explained by the first three axes (PCA-1 27.7 %,
PCA-2 19.7 %, PCA-3 15.3 %) inside the 95 % confidence interval.
PC-1 mainly explains the anti-correlation of TOC and $CaCO_3$. PC-2 shows the anti-correlation of K/I and S/I.
Positive loadings were observed for S/I, microcharcoal, macrocharcoal and $CaCO_3$. For PC-2, negative loadings
were observed for K/I, abundance of chlorite + I-S R1. PC-3 shows strong positive loadings (> 0.3) for $\delta^{13}C_{org}$,
Si/Al and Zr/Al.
Plotting PC-1 (y-axis) over PC-3 (x-axis) shows that the samples after the onset of the +ve CIE are grouped to
the top of the y-axis (more associated with S/I compared to K/I) and to the right of the x-axis (more associated
with primary minerals, phytoclasts, and higher Si/Al, Zr/Rb and Zr/Al) (Fig. 5).

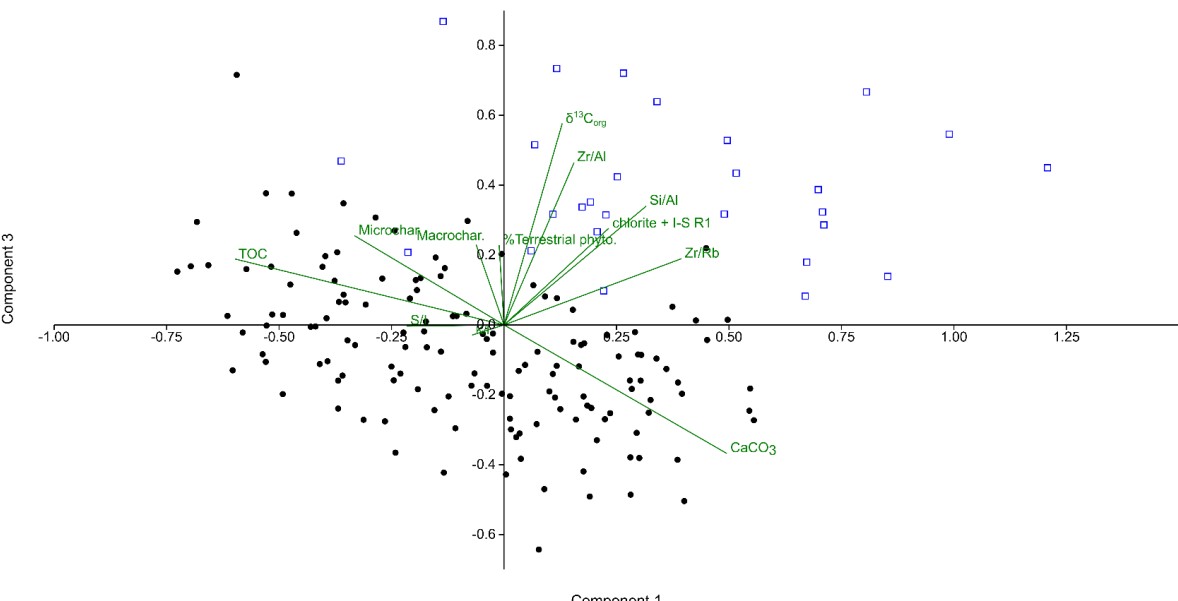

**Figure 5: PCA-analysis shows a distinctly different depositional signature before and after the onset of**
**the LPE +ve CIE in the Mochras core.** PCA plot of PC-1 and PC-3: all samples before the onset of the LPE
+ve CIE are marked in black closed circles and the samples after the onset of the LPE +ve CIE are marked in
blue open squares.

**1.5 Discussion**
Figure 2 provides the context for the LPE 'cooling event' at Mochras set within the background record. Shifts in
bulk $\delta^{18}O_{carb}$ are coeval to the $\delta^{13}C_{org}$ change to heavier isotopic values (~930 mbs) and reach a maximum in the
Margaritatus Zone (>1 ‰) (Ullmann *et al*., 2022). The bulk oxygen-isotope excursions of Mochras are affected
by diagenesis and are deemed unlikely to reflect environmental conditions (Ullmann *et al*., 2022). However,





oxygen isotope data from marine benthic and nektobenthic molluscs and brachiopods show heavier values
during the late Margaritatus Zone concurrent with a positive shift in $\delta^{13}C_{org}$, indicating cooling during the LPE
in the nearby Cleveland Basin (Robin Hood's Bay and Staithes) (Korte & Hesselbo, 2011) and this trend is also
observed in several European sections (e.g. Korte *et al.*, 2015). The duration of the +ve CIE has been estimated
as ~0.4–0.6 Myr in the Cardigan Bay Basin (Ruhl *et al.*, 2016; Storm *et al.*, 2020).

**1.5.1 Background sedimentological and environmental variations**
The Mochras succession shows metre-scale alternating TOC-enhanced and Ca-rich lithological couplets
(mudstone/limestone; Fig. 2). Previous assessments of the palaeoenvironmental signature of these TOC-
enhanced and Ca-rich couplets indicate strongly that the different depositional modes are driven by orbital
precession (Ruhl *et al.*, 2016; Hinnov *et al.*, 2018; Storm *et al.*, 2020; Hollaar *et al.*, 2021; Pieńkowski et al.,
2021). Precession driven changes in monsoonal strength have been suggested to influence the deposition and
preservation of TOC and carbonate in the Cardigan Bay Basin (Ruhl *et al.*, 2016), although the impact may have
been expressed, at least partially, by changes in strength of bottom currents in the seaway as a whole
(Pieńkowski *et al.*, 2021).
The preservation of primary carbonate is poor in the Mochras borehole, making it complex to determine in
detail the relative importance of carbonate producers for the bulk carbonate content (Ullmann *et al.*, 2022).
However, Early Jurassic, pelagic settings in the Tethys region often received abiotic fine grained carbonate from
shallow marine carbonate platforms (Weedon, 1986; Cobianchi & Picotti, 2001) and/or via carbonate producing
organisms (such as coccolithophores in zooplankton pellets) (Weedon, 1986; van de Schootbrugge *et al.*, 2005,
e.g. Weedon *et al.*, 2018). Coccolithophores are often poorly preserved and recrystallized (Weedon, 1986;
Weedon *et al.*, 2018). The organic matter found in the studied section of the Mochras borehole varies between
18 and 42% terrestrial phytoclasts (Fig. 3). Phytoclasts are common, but palynomorphs are relatively sparse and
poorly preserved. Marine amorphous organic matter is the main constituent in the present study of particulate
organic matter in unsieved macerated samples, in the interval studied here (951 – 918 mbs). Examination of
variations in the terrestrial/marine proportions of organic matter, shows no correspondence between the type of
organic matter and the TOC-enhanced or Ca-rich lithological alternations. However, previous research has
indicated that the percentage of terrestrial phytoclasts show precession forcing independent of the lithological
couplets (so out of phase with precession scale changes in Ca-TOC content) between 951 – 934 mbs in the
Mochras core (Hollaar *et al.*, 2021). Such orbital forcing of the terrestrial vs marine proportions of organic
matter were also found in Early Jurassic sediments of Dorset, and were similarly independent of the lithological
facies (Waterhouse, 1999). Terrestrial phytoclast content show a weak expression of long-eccentricity driven
variations in the section studied (Fig. 3).
Fossil charcoal makes up a substantial proportion of the organic fractions (11% of the terrestrial fraction) and
has previously been shown to vary considerably over long-eccentricity cycle 459 ± 1 peaking in abundance
during the phase of maximum eccentricity (Hollaar *et al.*, 2021). Microcharcoal also appears to be most
abundant during the maximum phase of the subsequent long eccentricity cycle 458 ± 1 (Fig. 3). Additionally,
K/I and S/I clay mineral ratios appear to alternate in response to long-eccentricity drivers (Fig. 3) up to 931 mbs



where the clay mineral signature changes. Between 951 and 930 mbs high K/I occurs during phases of low long
eccentricity suggesting an enhanced hydrological cycle (Hollaar *et al.*, 2021) with more intense weathering, and
enhanced fine grained terrestrial runoff to the marine record (Deconinck *et al.*, 2019). In contrast, phases of
maximum long-eccentricity appear to be smectite-rich, indicating seasonal rainfall, enhanced fire (Hollaar *et al.*,
2021) and thus periods of droughts, and lower terrestrial runoff and subsequent lower dilution (Deconinck *et al.*,
2019). Detrital elemental ratios increase accordingly during the smectite-rich phases, and are lower during
kaolinite-rich phases between 951 and 930 mbs. Detrital elemental ratios can be used to explore changes in
sediment composition (e.g. Thibault *et al.*, 2018; Hesselbo *et al.*, 2020b) and the similarity of the long-term
trend in Zr/Rb and Si/Al (Fig. 2) indicates that these elemental ratios reflect grainsize. The clay fraction (hosting
Al, and Rb (Chen *et al.*, 1999)), diminishes upwards, whereas the coarser silt to sand fraction (associated with Si
(Hesselbo *et al.*, 2020b) and Zr (Chen *et al.*, 2006)), increases upward (Fig. 3 and 4). The grainsize changes
inferred here reflect two overall coarsening upwards sequences (Fig. 3 and 4). These sequences may reflect
changes in clastic transport due to changes in the proximity to the shore/siliciclastic source, changes in runoff
due to a changing hydrological cycle, or accelerated bottom currents with greater carrying capacity of coarser
sediments.

### 394   1.5.2 Depositional and environmental changes before and after the LPE +ve CIE

The LPE +ve CIE begins around 930 mbs in the Mochras core and encompasses the remaining part of the
studied section (Fig. 3). We contrasted all the pre-CIE sediment signatures with those of the +ve CIE signatures
using principal components analysis which indicates a distinctly different sedimentary composition and
environmental signature before and after the onset of the +ve CIE in Mochras (Fig. 5).
Before the +ve CIE onset, the clay mineral assemblage shows alternating phases of smectite and kaolinite,
indicating pedogenic weathering. The relative abundance of the detrital clay types observed in the studied
interval have the potential to hold important palaeoclimatic information regarding the hydrological cycle and the
relative proportion of chemical weathering and physical erosion. Chemical weathering is enhanced in a high
humidity environment with relatively high temperatures and rainfall, when clays are formed in the first stages of
soil development. In the modern day, kaolinite is primarily formed in tropical soils, under year-round rainfall
and high temperatures (Thiry, 2000). Smectite also occurs in the tropics, but is more common in the subtropical
to Mediterranean regions, where humidity is still high, but periods of drought also occur (Thiry, 2000). Hence,
smectite forms predominantly in soil profiles under a warm and seasonally dry climate (Chamley, 1989; Raucsik
& Varga, 2008), and kaolinite in a year-round humid climate (Chamley, 1989; Ruffell *et al.*, 2002). Similarly,
alternating intervals of kaolinite and smectite dominance were observed for the Late Sinemurian (Munier *et al.*,
2021) and the Pliensbachian of Mochras (Deconinck *et al.*, 2019). The predominantly detrital character of these
clay minerals has been confirmed by TEM scanning of Pliensbachian smectite minerals, which revealed the
fleecy morphology and lack of overgrowth (Deconinck *et al.*, 2019). Therefore, the alternations of smectite and
kaolinite are interpreted to reflect palaeoclimatic signatures of a changing hydrological cycle, with a year-round
wet climate evidenced by high K/I ratios, and a more monsoon-like climate with seasonal rainfall with high S/I
(Deconinck *et al.*, 2019; Hollaar *et al.*, 2021; Munier *et al.*, 2021) (See Fig. 3). The intervals with a signal for



weaker seasons appears to correspond to phases of low eccentricity in the 405 kyr cycle, and signals of greater
seasonality with periods of high more pronounced eccentricity (Fig. 3) in the 405 kyr cycle.
Higher frequency cycles are not observed in the clay mineral ratios, with no precession or obliquity forcing
detected in the high-resolution part of the study 951 – 934 mbs (Hollaar *et al*., 2021) and no expression of the
100 kyr cycle in the record presented here. The formation of developed kaolinite-rich, and to a lesser extent
smectite-rich soil profiles, requires a steady landscape for many tens of thousands of years, although the ~1 Myr
timescale of Thiry (2000) seems excessive in our case given the clear expression of clay mineral changes
through long-eccentricity cycles. Also, the transportation and deposition of continental clays will occur after soil
formation and add further time between formation and final deposition (Chamley, 1989; Thiry, 2000). Thus,
there is likely to be a lag of the climatic signal observed in the marine sediments (Chamley, 1989; Thiry, 2000).
However, we note that high frequency climatic swings have been recorded in the clay mineral record in some
instances, such as in the Lower Cretaceous in SE Spain (Moiroud *et al*., 2012). The limestone-marl alternations
there are enhanced in smectite versus kaolinite and illite, respectively, reflecting precession scale swings from a
semi-arid to a tropical humid climate (Moiroud *et al*., 2012). Precession and higher frequency shifts in the clay
record are likely caused by fluctuations in runoff conditions rather than the formation of soils with a different
clay fraction.
Directly after the initial +ve CIE shift from 930–924 mbs (Phase 1 of Fig. 3) little seems to change, and the
system evidently continued to respond as before to the long eccentricity forcing, despite the predicted cooling
(Korte & Hesselbo, 2011; Korte *et al*., 2015; Gómez *et al*., 2016). However, from around 924 mbs up to the top
of the studied section (Phase 2 of Fig. 3) the clay mineral assemblage displays a distinctly different composition,
with kaolinite dominating especially the early part of phase 2 of the LPE (Fig. 3). At the same time there is an
enhancement of the primary minerals illite and chlorite, and I-S R1 (Fig. 3 and SI Fig. 1). Although an
enhancement in detrital kaolinite indicates an acceleration of the hydrological cycle, detrital kaolinite is dual in
origin and can also be derived from reworking of the primary source material (Deconinck *et al*., 2019). If the
climate is cooler, chemical weathering becomes less dominant and physical erosion of the bedrock becomes the
main detrital source of clay minerals. In the Cardigan Bay Basin, the bedrock of the surrounding Variscan
massifs (such as the Caledonian, Welsh and Irish massifs) were a likely source of these clays. In the Early
Jurassic of the NW Tethys region, mica-illite and chlorite bearing Lower Paleozoic mudrock were emergent
(Merriman, 2006; Deconinck *et al*., 2019), hence the enhancement of illite and chlorite likely indicates physical
erosion in the region surrounding the study site. Finally, authigenic clay particles could have been formed
during burial diagenesis. At temperatures between 60-70 °C smectite illitization occurs and I-S R1 is formed;
however, the high abundance of smectite in Mochras indicates limited burial diagenesis in the Mochras core
(Deconinck *et al*., 2019). Weak thermal diagenesis is confirmed for the Pliensbachian of Mochras, with $T_{max}$
between 421 °C and 434 °C (van de Schootbrugge et al., 2005). Therefore, I-S R1 in Mochras is interpreted to
be derived from chemical weathering of illite (Deconinck *et al*., 2019). The coeval increase of these primary
clay minerals, I-S R1 and kaolinite, indicate that during this period physical erosion dominated over soil
chemical weathering (Deconinck *et al*., 2019; Munier *et al*., 2021). This is similar to what is observed for the
latest Pliensbachian in Mochras (Deconinck *et al*., 2019).





Erosion of weathering profiles transports clay minerals (including kaolinite and smectite) to the marine realm. In
the ocean, the differential settling of kaolinite (near shore) and smectite (more distal) could occur based on the
morphology and size of clay particles (Thiry, 2000). However, comparison of long-term inferred regional sea
level changes from surrounding UK basins (Hesselbo, 2008) suggests that the relative proportions of smectite
and kaolinite are not influenced by changes in relative sea level in the Pliensbachian of Mochras (Deconinck *et*
*al*., 2019). On the assumption that the coarsening upward sequences at Mochras are indicative of relative sea
level change, it can also be argued that the proximity to shore did not impact the proportions of smectite and
kaolinite with enhanced smectite during 'proximal' deposition and enhanced kaolinite at times of more 'distal'
deposition (Fig. 3).
We suggest that the first phase of the LPE (Fig. 3, phase 1) was characterised by repeated periods of rainfall in a
seasonal climate forced by precession in which chemical weathering (smectite formation) dominated the
sedimentary signatures. This corresponds to maximum long-eccentricity and shows the same climatic signature
as during maximum eccentricity phases before the +ve CIE. This is then followed by a second phase (Fig. 3,
phase 2) where the climate is generally cooler, overall potentially more arid, but with rainfall throughout the
year over multiple precession cycles. This appears to have favoured deep physical erosion, owing to the
abundance of primary clay minerals, kaolinite and I-S R1. This interval corresponds to a minimum phase in the
405 kyr eccentricity based on Storm *et al*. (2020). This interpretation is further supported by decreasing and then
low microcharcoal abundance, pointing to suppression of fire activity at this time.
Two coarsening upward cycles that predate the onset of the +ve CIE and continue for a few metres after its
initiation, are present in the detrital elemental ratios (best expressed in Si/Al and Zr/Rb records) (Fig. 3 and 4),
and indicate a changing sediment influx over the studied interval. Previous study of the lithofacies of the
Mochras borehole has also shown the coarsening-upward sequences of 0.5–3 m thickness, which are observed to
be followed upwards by a thinner fining-upward succession (Pieńkowski *et al*., 2021). This reported fining-
upward part is not reflected in the elemental ratios of the two sequences shown in this study. Furthermore, the
coarsest phases of these sequences are approximately coeval with decreasing trends in the K/I ratio and
increasing trends in the S/I. This could indicate that periods of a strong monsoonal/seasonal climate (indicated
by S/I) brought coarser grained material to the basin, whereas periods of year-round humidity (K/I) are
associated with higher chemical weathering (low Si/Al). Therefore, these two coarsening upwards cycles appear
to link to increasing long-eccentricity. A similar mechanism has been inferred for the northern South China Sea
region in the Miocene, where coarser grained material is found during periods of a strong summer monsoon and
relatively lower chemical weathering (Clift *et al*., 2014). Present day studies show that bedrock erosion and
associated sediment transport is greater in areas with high seasonal contrast (Molnar, 2001; Molnar, 2004).
Hence, the Si/Al record also appears to reflect weathering and erosion conditions on land (Clift *et al*., 2014,
2020), driven by long-eccentricity modulated climate (SI Fig.5). However, other scenarios that would influence
the grain size on this time scale cannot be dismissed and include changes in proximity to siliciclastic source, or
changes in sediment transport via bottom water currents.
Changes in bottom water current strength and direction likely affected the depositional site of the Mochras core
(Pieńkowski *et al*., 2021) although there is as yet no consensus on the processes that likely controlled these
palaeoceanographic parameters. An early phase of regional tectonic updoming of the North Sea disrupted the



circulation in the N-S Laurasian Seaway (including the Viking Corridor) and therefore diminished the
connectivity between western Tethys and the Boreal realm, hypothetically reducing poleward heat transport
from the tropics (Korte *et al*., 2015). This mechanism has also been argued to explain the later cooling observed
in NW Europe during the transition of the warmer Toarcian to the cooler Aalenian and Bajocian (Korte *et al*.,
2015). Late Pliensbachian occlusion of the Viking Corridor is supported by the provincialism of marine faunas
at this time, showing a distinct Euro-Boreal province and a Mediterranean province (Dera *et al*., 2011b). During
the Toarcian a northward expansion of invertebrate faunal species has been found (Schweigert, 2005; Zakharov
*et al*., 2006; Bourillot et al., 2008; Nikitenko, 2008), indicating a northward (warmer) flow through the Viking
corridor (Korte *et al*., 2015). More recently, a southward expansion of Arctic dinoflagellates into the Viking
Corridor was suggested for the termination of the T-OAE (van de Schootbrugge *et al*., 2019), which is in
agreement with a N to S flow through the Viking Corridor suggested by numerical models (Bjerrum *et al*., 2001;
Dera & Donnadieu, 2012; Ruvalcaba Baroni *et al*. 2018) and sparse Nd-isotopes (Dera *et al*., 2009).
Over the European Epicontinental Shelf (EES), and the Tethys as a whole, a clockwise circular gyre likely
brought oxygenated warm Tethyan waters to the southwest shelf, with a progressively weaker north and
eastward flow due to rough bathymetry and substantial islands palaeogeography (Ruvalcaba Baroni *et al*.,
2018). This predominantly surface flow is modelled to have extended to shelfal sea floor depths. Only
episodically might nutrient-rich Boreal waters have penetrated south onto the EES in these coupled ocean-
atmosphere GCM model scenarios (Dera & Donnadieu, 2012). The modelling also suggests – counter-
intuitively – that the clockwise surface gyre of the Tethys extended further northwards and impacted the EES
more effectively when the Hispanic corridor was more open.
An alternative bottom current configuration was discussed for Mochras specifically wherein changes in north-
to-south current strength (cf. Bjerrum et al., 2001) are proposed for the changes in grainsize and siliciclastic
versus clay content via contour currents (Pieńkowski *et al*., 2021). A strong flow from the cooler and shallow
boreal waters is hypothesized to have brought a coarser grainsize fraction in suspension and as bedload, which
was then deposited in the Cardigan Bay Basin while flowing to the deeper and warmer waters of the peri-Tethys
(Pieńkowski *et al*., 2021). Times of a strong north to south current are proposed to be associated with more
oxygenated bottom waters (Pieńkowski *et al*., 2021). In contrast, when the north to south current became
weaker, less coarse material will have been carried in suspension and as bedload, and a relatively higher clay
proportion will have been deposited in the Cardigan Bay Basin (Pieńkowski *et al*., 2021). In this scenario, times
of sluggish currents are associated with low bottom water oxygenation (Pieńkowski *et al*., 2021) and thus
climate forcing of current strength could explain the deposition of alternating coarser and finer fractions in the
Mochras borehole (Pieńkowski *et al*., 2021).
Our research suggests that orbital cycles both before and during the onset of the +ve CIE have a significant
influence on seasonality and hydrology, affecting both fire regimes and sediment depositional character. Further
research is required to consider how long-eccentricity and obliquity cycles might interact with north-south flow
in the Cardigan Bay Basin and circulation processes. What is clear is that orbital cycles have impact on
terrestrial processes in the terrestrial sediment source areas (Hollaar *et al*., 2021) and led to differences in
deposition within the marine sediments in Mochras core (Ruhl *et al*., 2016; Pieńkowski *et al*., 2021). Our data
indicate that periods of coarser sediment deposition correspond to periods that include more seasonal climates



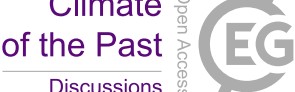

before the onset of the +ve CIE (low kaolinite), which is in line with the hypothesized grainsize changes caused
by contour currents (Pieńkowski *et al*., 2021). However, after the onset of the +ve CIE, although we suggest that
the chemical weathering rate decreased, enhanced runoff and physical erosion are indicated by a peak in primary
clay minerals and K/I. Enhanced runoff could be expected to impact the thermohaline contour currents (Dera &
Donnadieu, 2012). Simultaneously, an increasingly cold climate (as indicated by enhanced physical erosion over
chemical weathering) indicates a boreal influence. It remains to be determined to what extent orbital cycles
might have the power to influence ocean circulation in the basin.
Relatively coarse sediments in the Late Pliensbachian have also been related to shallower sediment deposition in
UK basins (Hesselbo & Jenkyns, 1998; Hesselbo, 2008; Korte & Hesselbo, 2011). These regressive facies may
have been caused by an early phase of North Sea doming (Korte & Hesselbo, 2011). Sequence stratigraphy of
the Lower Jurassic of the Wessex, Cleveland and Hebrides basins (Hesselbo & Jenkyns, 1998; Hesselbo, 2008)
shows relative sea level changes and sand influxes in the late Margaritatus Zone in the studied basins.
Noteworthy in the Mochras borehole are phases of low $\delta^{18}O$ of macrofossils which seem to correspond to high
phases of macrofossil wood concomitant with low sea level, suggesting a possible control of relative sea level
on the oxygen-isotope record and the source of detrital material (Ullmann *et al*., 2022). The broad spatial
distribution of these basins suggests that associated regression and/or sediment influx is of at least regional scale
(Hesselbo, 2008). The results presented here fall within this phase of regression (Hesselbo & Jenkyns, 1998;
Hesselbo, 2008).
In the context of North Sea doming as a possible cause of the Late Pliensbachian cooling, these facies can be
interpreted to represent shallowing upward facies in a shallower system, or deep water system receiving coarser
sediment input. The doming is hypothesized to have minimized or prohibited southward flow of cooler waters
from the Boreal and northward flow from warmer waters from the Mediterranean area (Korte *et al*., 2015). The
Mochras borehole is situated on the southwestern flank of the dome and would have been cut-off from the
northern parts of the Laurasian Seaway, including the Hebrides Basin and Cleveland Basin (Korte *et al*., 2015).
This change in seaway circulation could have impacted the source area of the detrital sediments in the Mochras
borehole and brought the shallow shoreface facies closer to the borehole site.
Doming of the North Sea area would have led to greater radial spread of nearshore facies; however, owing to the
strong eccentricity forcing that we interpret here, an additional factor that is influenced by the seasonal
distribution of insolation forced by orbital cyclicity needs to be included. The Cardigan Bay Basin (Mochras) is
positioned about 290 km to the SW of the Cleveland Basin and at a similar latitude, but to the W of the Wessex
Basin (Ziegler, 1990; Torsvik & Cocks, 2017), and is therefore expected to be impacted by the regional changes
in sea level and/or sediment flux. In the Late Pliensbachian of the Cleveland Basin, the detrital ratios of Si/Al,
Zr/Al and Zr/Rb show similar coarsening upward sequences, which have been interpreted to reflect changes in
riverine transport of siliciclastic grains and grainsize (Thibault *et al*., 2018). The inferred changes in sea-level in
the Cleveland Basin occur at a 100 kyr pacing (Huang & Hesselbo, 2014; Hesselbo *et al*., 2020b), potentially
linking the regression cycles to short eccentricity (Huang *et al*., 2010 and refs therein) and long-eccentricity
(Thibault *et al*., 2018). This would mean that eccentricity driven changes in inferred sea level change could be
linked to glacioeustatic cycles during these times (Brandt, 1986; Suan *et al*., 2010; Korte & Hesselbo, 2011;
Krencker *et al*., 2019; Ruebsam *et al*., 2019, 2020b; Ruebsam & Schwark, 2021; Ruebsam & Al-Husseini,



2021). Glacioeustatic sea level changes are discussed for the Early Jurassic and Middle Jurassic (Krencker *et al.*,
2019; Bodin *et al.*, 2020; Ruebsam & Schwark, 2021; Nordt *et al.*, 2022). A recent study on the rapid
transgression observed at the Pliensbachian–Toarcian boundary, ruled out other mechanisms that could force sea
level at this time scale, such as aquifer-eustacy, and show that glacioeustatic changes in sea level are a likely
possibility at times in the Early Jurassic (Krencker *et al.*, 2019). Therefore, our findings overall provide support
the episodic occurrence of continental ice at the poles (Brandt, 1986; Price, 1999; Suan *et al.*, 2010; Korte &
Hesselbo, 2011; Korte *et al.*, 2015; Bougeault *et al.*, 2017; Krencker *et al.*, 2019; Ruebsam *et al.*, 2019, 2020a,
2020b; Ruebsam & Schwark, 2021; Ruebsam & Al-Husseini, 2021).

**1.6 Conclusions**
The terrestrial environment adjacent to the Cardigan Bay Basin was strongly influenced by both orbitally driven
climate forcings and the Late Pliensbachian Cooling Event (LPE). Long-eccentricity forcing remained strong
both prior to and during the LPE. Prior to the LPE, eccentricity-driven shifts in maximum seasonality influence
the degree of chemical weathering (S/I *vs* K/I), sediment flux to the basin (Si/Al), and fire activity. As maximum
precessional seasonality decreases with reduced 405 kyr eccentricity, the year-round relatively cool and wet
climate extended over multiple precession cyles drove significant erosion of bedrock on emergent land surfaces
as evidenced by high bedrock-derived mineral content, high K/I and I-S R1. Therefore, both the Milankovitch
forcings and larger climatic shifts operate in tandem to drive changes in the terrestrial environment.

**Data availability**: Supplementary data is available at the National Geoscience Data Centre at Keyworth
(NGDC) at (doi to be added) for the interval 934 – 918 mbs. All data presented for the interval 951 – 934 mbs is
available at the National Geoscience Data Centre at Keyworth (NGDC) at https://doi.org/10.5285/d6b7c567-
49f0-44c7-a94c-e82fa17ff98e (Hollaar *et al.*, 2021). The full Mochras XRF dataset is in Damaschke *et al.*

594   (2021).

**Author contribution**: CMB, SPH and TPH designed the research. TPH conducted the laboratory
measurements, with JFD contributing to the XRD-measurements and MD, CU and ML to the XRF-
measurements. TPH, CMB and SPH wrote the manuscript, with contributions of all authors.
**Competing interests**: The authors declare that they have no conflict of interest.
**Acknowledgements**: This is a contribution to the JET project funded by the Natural Environment Research
Council (NERC) (grant number NE/N018508/1). SPH, CMB, JFD, CU, ML and TPH, acknowledge funding
from the International Continental Scientific Drilling Program (ICDP) and TPH acknowledges funding from the
University of Exeter. We thank the British Geological Survey (BGS), especially James Riding, Scott Renshaw
for facilitating access to the Mochras core. Also, Simon Wylde and Charles Gowing for their contribution to the
XRF scanning and discussion on the results. We further thank Chris Mitchell for help with the TOC and $\delta^{13}C_{org}$
analyses. Finally, we thank Ludovic Bruneau for technical assistance with the XRD-analysis.





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
