# Peer review of "Environmental changes during the onset of the Late Pliensbachian 1"

_Climate of the Past, 2022_

## Author Response (AR1)

**Review 1**

The authors present a high-resolution geochemical, palynological, and clay mineral assemblage dataset from the Upper Pliensbachian of the Mochras core. This dataset is used to discuss the factors at the origin of the Late Pliensbachian cooling event. Overall, this study brings new and interesting data and discussion about an event that could have seen the development of polar glaciation during the Mesozoic, and I therefore recommend it for publication after moderate revisions. Those suggested revisions are linked to my two main comments about this manuscript; Firstly, the structure of the discussion; Secondly, the claim of an earliest onset of the North Sea doming compared to a Late Toarcian – Early Aalenian onset as generally described in the literature

The Chapter 1.5.2 is rather long and confusing as the main guideline of the discussion is not straightforward. I would recommend to separate the discussion about the origin of the clay mineral and their assemblage variation in a separated subchapter that should appear at the beginning of the discussion. Further subdivision of this chapter might also help its readability.

In the literature, the onset of the North Sea Doming is said to occur around the Toarcian-Aalenian transition (e.g., Underhill & Partington, 1993). The claim of an earlier onset of this tectonic event can be tracked to the discussion part of Korte et al. (2015) as a putative explanation for the Late Pliensbachian cooling in light of the Aalenian cooling example. These authors were using the argument of regressive facies in the upper Pliensbachian of the North Sea region to substantiate their proposal of an earlier onset. However, Late Pliensbachian regression is a worldwide phenomenon observed in far away regions such as North Africa, the Sverdrup Basin in Canada, the Neuquen Basin in Argentina, or the Arabian Plate. Using this regressive trend as an argumentation for an earlier start of the North Sea Doming is therefore not warranted. I would suggest to tone this hypothesis down throughout the manuscript and clearly emphasise its limitation. This applies notably for the abstract, as well as the part in lines 492–494, where sentences such as "An early phase of regional tectonic updoming of the North Sea disrupted the circulation in the N-S Laurasian Seaway (including the Viking Corridor) and therefore diminished the connectivity between western Tethys and the Boreal realm…" give the impression that this putative early updoming phase is a well-established fact.

In general, I actually don't think that there is a need to evoke such tectonic phase. The global Late Pliensbachian sea-level low stand might on its own account for poor connectivity between western Tethys and the Boreal realm without having to invoke a regional uplift in the North Sea. Hence, instead of using "North Sea doming", I would for instance adopt a more neutral position and favour terms like "shallowing in the North Sea and Viking corridor".

Below, I also list some line specific comments. Overall, I enjoyed reading this manuscript and I'm happy to see that Mesozoic cooling events are also of interests for other research groups.

Stéphane Bodin

Dear Stéphane Bodin,

Thank you for your constructive feedback for our manuscript. Your suggestions will be very helpful in refining the manuscript and improving the structure. We will modify the discussion and incorporated subheadings to increase the readability as you suggest. The discussion on clay mineralogy has been moved to the first part of the discussion (Line 423 onwards), directly after the clay ratios are first mentioned.

With regard to the North Sea Doming, we have carefully reword the parts of the manuscript that mention this. The North Sea Dome is a topographical structure that has been inferred to be present in the North Sea since the upper Toarcian with hints that it might have existed earlier (e.g. see recent summary in Archer et al. (2019)). Regressive facies characterise various points along the Laurasian Seaway including further north in the Viking Corridor (e.g. Folkstad and Steel, 2023). We hypothesize in line with Korte et al. (2015) that a sea level low stand – which we agree is very widely recorded for the Late Pliensbachian – would have led to the relative emergence of any regional topographic feature at the time. The important point is that the regionally observed regression during the Late Pliensbachian implies some obstruction of flow within the Viking Corridor.

Specific comments:

Line 35: Problem in the numeration of the chapter. The Introduction should by numbered under "1" and "1.1"? Same remark for all the main chapter of this manuscript.

Yes, thank you for noticing this, this have changed the numbering of the sections.

Line 44: You could here make reference to Bodin et al. (2023) which further confirms the temporal correlation between the LPE and a long-term sea-level low stand.

Thank you for the reference, this has been added in.

Adding a figure (maybe in Figure 1) showing the Upper Pliensbachian of the Mochras core and its d13C record, and highlighting in it the high-resolution studied part would help to better contextualize the here-presented data.

Thank you for the suggestion. This will indeed be important to give a contextual overview to the reader of the Pliensbachian of Mochras. We have merged the original figure 4 with your suggestions, into a new figure 1. We provide the stratigraphic overview of Mochras of the Pliensbachian, with the d13Corg record from Storm et al. (2020) and the orbital filters, next to the Laskar 405 kyr metronome. We have highlighted our study area so it is clear immediately what the context is.

Line 336: Shouldn't that be a reference to Fig 4 instead of Fig 2 as written?

Yes, thank you and text has been adapted.

Lines 357–359: "pelagic settings in the Tethys region often received abiotic fine grained carbonate […] via carbonate producing organisms (such as coccolithophores in zooplankton pellets)". Calcareous nannoplankton production was very limited during the Jurassic and likely not at the origin of limestone-marl alternations. These latter are best explained by the cyclic export of shallow marine carbonates, as deduced from the disappearance of limestone-marl alternations during time of neritic carbonate factory collapse (e.g. Krencker et al., 2020)

Thank you for the new reference, we have incorporated this. Although the quantity of the origin of nannoplankton versus carbonate platform carbonate is largely unknown for the Early Jurassic of the UK, there are several studies that indicate that the carbonate in the limestone – mudstone alternations in the UK consists of nannoplankton (e.g. Weedon et al., 2019). Furthermore, a recent study from Slater et al. (2022) has shown the presence of cryptic nannoplankton preserved only as impressions in organic matter for the Toarcian of the UK, Germany, Japan, and New Zealand. Therefore, we think it is justified to say that carbonate in the Early Jurassic can be derived from both carbonate sources.

Lines 388–392: "The grainsize changes inferred here reflect two overall coarsening upwards sequences (Fig. 3 and 4). These sequences may reflect changes in clastic transport due to changes in the proximity to the shore/siliciclastic source, changes in runoff due to a changing hydrological cycle, or accelerated bottom currents with greater carrying capacity of coarser sediments". Could this grainsize change also reflect weathering intensity, with parent rocks being weathered down toward finer grain size during more intense weathering periods as indicated by high K/I and low S/I, and vice versa?

Yes, this could also be one of the possibilities, thank you for suggesting this. We have added this in the sentence (L452-455). Enhanced K/I could be indicative of enhanced runoff/intensified weathering, which could have led to a finer sediment fraction.

Lines 399–417: This discussion about the interpretation of clay mineral assemblages change should appear earlier in the text as the K/I and S/I ratio are already used in chapter 1.5.1.

Thank you. We will move this section to the first part of the discussion (L398 – 414) after the clays are first mentioned in the discussion.

Lines 514–515: "siliciclastic versus clay content". Change "siliciclastic" for "silt" as the clay in the core are also siliciclastics.

Thank you, this have changed this to silt/sand.

Lines 540–541: "These regressive facies may have been caused by an early phase of North Sea doming". As already stated earlier, this regressive trend is seen on a global scale and can therefore not be considered as a footprint of the North Sea Doming.

Yes, thank you. We have changed our wording, so it reflects the likely global regression and a regional topography that is impacted by relative sea-level fall of at least regional extent.

References cited:

Bodin, S., Fantasia, A., Krencker, F.-N., Nebsbjerg, B., Christiansen, L., Andrieu, S., 2023. More gaps than record! A new look at the Pliensbachian/Toarcian boundary event guided by coupled chemo-sequence stratigraphy. Palaeogeography, Palaeoclimatology, Palaeoecology 610, 111344.

Krencker, F.-N., Fantasia, A., Danisch, J., Martindale, R., Kabiri, L., El Ouali, M., Bodin, S., 2020. Two-phased collapse of the shallow-water carbonate factory during the late Pliensbachian–Toarcian driven by changing climate and enhanced continental weathering in the Northwestern Gondwana Margin. Earth-Science Reviews 208, 103254.

Underhill, J.R., Partington, M.A., 1993. Jurassic thermal doming and deflation in the North Sea: implications of the sequence stratigraphic evidence. Geological Society, London, Petroleum Geology Conference series 4, 337–345.

Additional references cited:

Archer, S. G., Steel, R. J., Mellere, D., Blackwood, S., & Cullen, B. (2019). Response of Middle Jurassic shallow-marine environments to syn-depositional block tilting: Isles of Skye and Raasay, NW Scotland. Scottish Journal of Geology 55, p. 35–68.

Folkestad, A. & Steel, R. J. (2023). A new interpretation for the Pliensbachian Cook Formation (northern North Sea) as north–south prograding tidal deltas and shelf ridges in the Early Jurassic Seaway: new model of linkage to the Norwegian Sea. In: Rossi, V. M., Longhitano, S., Olariu, C. and Chiocci, F. (eds) Straits and Seaways: Controls, Processes and Implications in Modern and Ancient Systems. Geological Society, London, Special Publications, 523, https://doi.org/10.1144/SP523-2021-75

Slater, S. M., Bown, P., Twitchett, R. J., Danise, S., & Vajda, V. (2022). Global record of "ghost" nannofossils reveals plankton resilience to high CO2 and warming. Science, 376(6595), 853-856.

Weedon, G. P., Page, K. N., & Jenkyns, H. C. (2019). Cyclostratigraphy, stratigraphic gaps and the duration of the Hettangian Stage (Jurassic): insights from the Blue Lias Formation of southern Britain. Geological Magazine, 156, 1469-1509.

**Review 2**

Review for Hollaar et al. - Environmental changes during the onset of the Late Pliensbachian Event (Early Jurassic) in the Mochras Borehole, Cardigan Bay Basin, NW Wales.

The work by Hollaar et al. presented detailed data on late Pliensbachian strata from the Mochras Core. Data provide detailed insights into environmental (oceanographic and continental weathering) during a period of major environmental change. The paper is well written and logically structures. Interpretation of the data are sound and proof and supported by the data. I basically agree with the interpretations by the authors.

However, I'd like to point out one issue that might require some attention. Much of the oceanographic changes reconstructed for late Pliensbachian times is linked to updoming in the North Sea area. Updoming can explain the regional development of a regressive facies and to major changes in current systems across the shallow shelf sea. According to my understanding and what have read in the works by Underhill and coauthors, North Sea dominig occurred in the late Toarcian to early Aalenian (e.g., Underhill and Partington, 1993; GSL – Petr.Geol. Conf. 4, 337-345). Korte et al. (2015) argued that dominig in the late Toarcian was one parameter controlling shelf currents and heat transport across the shallow shelf. Are you sure that the same model can be applied to the late Pliensbachian? To my best knowledge, the works quoted by Korte et al. (2015) provide no evidence for doming during the late Pliensbachian. Maybe the authors can discuss this issue more detailed.

Some specific comments below. Hope you consider the comments constructive and helpful.

Best regards,

Wolfgang Ruebsam

Dear Wolfgang Ruebsam,

Thank you for your extensive and constructive feedback of our manuscript. Your suggestions will help us improve the current manuscript.

As we have responded to reviewer 1, we have changed and refine our wording when we talk about North Sea doming in the manuscript. We no longer call it 'doming', but the North Sea dome structure, which is an important feature of the North Sea topography since the upper Toarcian, with hints that it might have existed earlier (e.g. see recent summary in Archer et al. (2019)).  Regressive facies characterise various points along the Laurasian Seaway including further north in the Viking Corridor (e.g. Folkstad and Steel, 2023). We hypothesize in line with Korte et al. (2015) that a sea level low stand – which we agree is very widely recorded for the Late Pliensbachian – would have led to the relative emergence of any regional topographic feature at the time. The important point is that the regionally observed regression during the Late Pliensbachian implies some obstruction of flow within the Viking Corridor.

**Lines 36-37:** Fully agree that marked climate changes occurred throughout the Early Jurassic. However, the view of an overall warm and high pCO2 Early Jurassic word has been challenged by several works. It is more likely that Jurassic climate shifted between cold and warm phases, including icehouse periods (e.g., Dera et al., 2011; Korte and Hesselbo, 2011; Krencker et al., 2019; Ruebsam and Schwark, 2021). The work by McElwain et al. (2005) provides stomata-based pCO2 estimates for the early Toarcian only. This work provides no information on CO2 levels in the Pliensbachian. This work further attests to contrasting CO2 levels in the early Toarcian. Thus, quoting to McElwain et al. (2005) and saying that the Early Jurassic was an overall warm and high pCO2 world is not correct.

Thank you for this feedback. We fully agree that current research shows that the Early Jurassic climate oscillated between warming and hyperthermal events, and relative cooling snaps (e.g. Storm et al., 2020 for compilation). However, the literature does indicate a consensus that the climate between these cold snaps was warmer than today. Since McElwain et al. (2005) only sampled from the Toarcian, we have included Steinthorsdottir & Vajda (2015) for Pliensbachian $pCO_2$, Korte et al. (2015) for Late Pliensbachian to Middle Jurassic temperature reconstruction and Robinson et al. (2016) for Sinemurian and Pliensbachian sea surface temperatures. In addition, have added '[…in an overall warm**er than present**…]' to be more exact.

**Lines 60-63:** Updoming in the North Sea region will have impacted current systems at the northwestern West Tethys shelf. This area was a very shallow seaway (shelf area) and I'm not convinced that changes in water circulation at this shallow shelf will have impacted global ocean circulation. Changes in the thermohaline circulation may have occurred in relation to global-scale tectonic changes (continent configuration). The work by Bjerrum et al. (2001) further attests to the presence of southwards-directed current system throughout the Viking Corridor. Thus, there no proof that warm Tethyan current transported warm water massed towards polar latitudes via this narrow seaway. The current system indicated in figure 1 (red arrow) is speculative. On the contrary, there is robust evidence (e.g., 18O data; modeling) that a southwards-directed Arctic current system transported low-saline cooler water masses to the northwestern West Tethys shelf via the Viking Corridor (Bjerrum et al., 2001; Dera and Donnadieu, 2012).

Thank you for the feedback. To address the first part of your comment: we will replace 'North Sea updoming' with 'sea level low stand'.

To answer the second part of your comment: evidence of a northward current is summarized and presented in Korte et al. (2015) and the cut-off of the northward flow has been suggested as an alternative cooling mechanism. In our manuscript we present the evidence for both current directions. There is not a consensus for current directions which may anyway have changed through time (Ruvalcaba Baroni et al. 2018) even if the predominant current may have been from north to south. We also note that some of the seaway straits may have been deeper than has been previously appreciated (Pienkowski et al. 2021). Notwithstanding, this topic is only summarised here to illustrate the different mechanisms that have been proposed for cooling at that time. Later in the manuscript we summarise the published relevant literature as follows:

L494 – 504 ' [….] hypothetically reducing poleward heat transport from the tropics (Korte et al., 2015). This mechanism has also been argued to explain the later cooling

observed in NW Europe during the transition of the warmer Toarcian to the cooler Aalenian and Bajocian (Korte et al., 2015). Late Pliensbachian occlusion of the Viking Corridor is supported by the provincialism of marine faunas at this time, showing a distinct Euro-Boreal province and a Mediterranean province (Dera et al., 2011b). During the Toarcian a northward expansion of invertebrate faunal species has been found (Schweigert, 2005; Zakharov et al., 2006; Bourillot et al., 2008; Nikitenko, 2008), indicating a northward (warmer) flow through the Viking corridor (Korte et al., 2015). More recently, a southward expansion of Arctic dinoflagellates into the Viking Corridor was suggested for the termination of the T-OAE (van de Schootbrugge et al., 2019), which is in agreement with a N to S flow through the Viking Corridor suggested by numerical models (Bjerrum et al., 2001; Dera & Donnadieu, 2012; Ruvalcaba Baroni et al. 2018) and sparse Nd-isotopes (Dera et al., 2009).'

**Method part - lines136 and following:** Detailed data on the Mochras Core geochemistry have been published previously: Ruhl et al., 2016 – XRF; Strom et al., 2020 – TOC, H, 13Corg; can the author please clarify if all data presented here (TOC, 13Corg, XRF) have been newly generated in this study? If data were taken from previous studies, those works must be quoted.

These data have been newly generated. L137-139: we write that the **new** samples (all samples between 918-934 mbs) complement a set of samples published in Hollaar et al. (2021), samples between 934 – 951 mbs. No data described in the methods/results are from any other publication. We have reviewed the manuscript to make this clear throughout.

**Figure 4:** This figure nicely defines the late Pliensbachian +veCIE. I think most (or all?) of the data shown here have been generated in previous works and are not part of this study. Thus, this figure should be shown in the introduction part and not in the results. Showing the figure in the introduction may give the reader a good impression of the late Pliensbachian events, as recorded in the sediments of the Mochras Core.

Yes. This figure is included to show context on the Late Pliensbachian +ve CIE in the Mochras borehole and does not include any new data. We moved the figure to the introduction part of the manuscript (now part of figure 1). More context has been added at the same time.

**Figure 3:** Figure 3 should be shown after figure 4, as the Pliensbachian events are defined in the latter. Moreover, figure 3 shows a lot of data interpretation, which should not be part of the results section.

Figure 3 is a compilation figure that does include data interpretation, but also contains a synthesis of all results that are not presented in any other figures, and is thus important to refer to during the result section and discussion section. However, Figure 4 has been moved forward (see above).

**Lines 355-372:** Here changes in the composition and nature of the sedimentary organic matter are described on the basis of palynological data. These data could be compared with HI/OI data that were presented in Storm et al. (2020). Integration of Rock Eval (or HAWK) data may allow assessing changes in the preservation of marine organic matter.

We chose not to make this comparison on the basis that Storm et al. (2020, p.3977) stated: "The HI values recorded in Mochras are likely highly compromised as a result of aerobic bacterial degradation of initially hydrogen-rich marine organic matter and are therefore not indicative for the primary source of the organic matter." Storm et al. based their understanding on study of the organic petrology.

**Lines 355-372:** Could changes in grain size (and Zr/Rb, Si/Al) also be affected by sea-level variations? 4[th]order sea-level cycles have been related to long-eccentricity forcing. Could be interesting to explore this aspect.

Yes, definitely, this is outlined in L592-611.

**Lines 448-449:** Tmax values of 421-434°C indicate that the sediments (and the organic matter) may have reached the early oil window and experienced a burial temperature of at least 60°C. Thus, the thermal maturity/diagenesis should be classified as weak-moderate.

Thank you, this has been adapted to
'Weak-moderate thermal diagenesis is confirmed for the Pliensbachian of Mochras, with Tmax from pyrolysis analysis between 421 °C and 434 °C (van de Schootbrugge et al., 2005; Storm et al., 2020).'

**Line 492 and following:** Korte et al (2015) explained that North Sea doming and uplift occurred in the late Toarcian (relevant works were quoted in Korte et al.). Is there any data that support an early updoming in the Pliensbachian?

The late Pliensbachian record a long-term sea-level lowstand (e.g., Haq, 2018). A low eustatic sea level will have narrowed small ocean gateways (such as the Viking Corridor) and thereby impacted current systems and faunal realm. This aspect should be added to the discussion. In lines 501 and following it is explained how a sea level highstand in the Bifrons Zone could have terminated anoxia in the European Basin System. This indicates that eustatic sea level changes strongly impacted oceanographic conditions at this shallow shelf sea.

Yes, thank you for this feedback. We have changed the wording so it is clear that an at least supra-regional sea level low stand would have led to the occlusion of the Viking Corridor.

**Lines 494-495:** As pointed out before, the current directions in at the northwestern Tethys shelf remains debated. There is strong evidence for a southwards directed Arctic current and (to my best knowledge) not strong evidence for a northward directed current through the Viking corridor (e.g., Bjerrum et al., 2001; Dera and Donnadieu, 2012).

Thank you. We explore the option of cooling caused by an obstruction of the warm north directed current as suggested by the work of Korte et al. (2015). Korte et al. (2015) present evidence for a northward current based on:

L498-501: 'During the Toarcian a northward expansion of invertebrate faunal species has been found (Schweigert, 2005; Zakharov et al., 2006; Bourillot et al., 2008; Nikitenko, 2008), indicating a northward (warmer) flow through the Viking corridor (Korte et al., 2015).'

And northward currents are derived from: Damborenea, S. E., Echevarrı´a, J. & Ros-Franch, S. Southern Hemisphere Palaeobiogeography of Triassic-Jurassic Marine Bivalves (Springer, 2013).

Other parts of our manuscript also summarize the evidence for a southward current (as specified in previous answer). And we build on this research in the manuscript as well when we discuss ocean circulation.

**Lines 510-512:** Correct, but s is highly speculative if the Hispanic Corridor efficiently connected the Panthalassic and the Tethys Ocean and thereby impacted global ocean circulation pattern during the Pliensbachian-Toarcian.

We have added a sentence: 'The timing of the opening of the Hispanic corridor is debated and varies from the Hettangian to Pliensbachian (Aberhan, 2000; Porter et al., 2013; Sha, 2019).'

**Lines 540-541:** As mentioned earlier, the late Pliensbachian records a global eustatic sea level lowstand. It is unlikely that the doming was the major factor causing the development of a regressive facies.

Yes, thank you. We have modified the sentence appropriately, emphasising that at least regional relative sea-level fall (possibly global) caused widespread regressive facies at many points in the seaway, especially where local topographic features existed.

Additional references cited:

Aberhan, M. (2001). Bivalve palaeobiogeography and the Hispanic Corridor: time of opening and effectiveness of a proto-Atlantic seaway. Palaeogeography, Palaeoclimatology, Palaeoecology, 165(3-4), 375-394.

Archer, S. G., Steel, R. J., Mellere, D., Blackwood, S., & Cullen, B. (2019). Response of Middle Jurassic shallow-marine environments to syn-depositional block tilting: Isles of Skye and Raasay, NW Scotland.  Scottish Journal of Geology 55, p. 35–68.

Folkestad, A. & Steel, R. J. (2023). A new interpretation for the Pliensbachian Cook Formation (northern North Sea) as north–south prograding tidal deltas and shelf ridges in the Early Jurassic Seaway: new model of linkage to the Norwegian Sea.  In: Rossi, V. M., Longhitano, S., Olariu, C. and Chiocci, F. (eds) Straits and Seaways: Controls, Processes and Implications in Modern and Ancient Systems. Geological Society, London, Special Publications, 523, https://doi.org/10.1144/SP523-2021-75

Porter, S. J., Selby, D., Suzuki, K., & Gröcke, D. (2013). Opening of a trans-Pangaean marine corridor during the Early Jurassic: Insights from osmium isotopes across the Sinemurian–Pliensbachian GSSP, Robin Hood's Bay, UK. Palaeogeography, Palaeoclimatology, Palaeoecology, 375, 50-58.

Robinson, S. A., Ruhl, M., Astley, D. L., Naafs, B. D. A., Farnsworth, A. J., Bown, P. R., ... & Markwick, P. J. (2017). Early Jurassic North Atlantic sea-surface temperatures from TEX 86 palaeothermometry. Sedimentology, 64(1), 215-230.

Sha, J. (2019). Opening time of the Hispanic Corridor and migration patterns of pan-tropical cosmopolitan Jurassic pectinid and ostreid bivalves. Palaeogeography, Palaeoclimatology, Palaeoecology, 515, 34-46.